# OpenReview forum: "Adaptive Self-Supervised Learning Strategies for Dynamic On-Device LLM Personalization"
_ICLR.cc/2025/Conference — Submitted to ICLR 2025_

### Official Review · Reviewer_akyN · 2024-10-28

**Soundness:** 2
**Presentation:** 2
**Contribution:** 2
**Rating:** 5
**Confidence:** 3

**Summary:**

This paper introduces Adaptive Self-Supervised Learning Strategies (ASLS) to address the challenge of personalizing large language models (LLMs) for individual users, particularly in resource-constrained on-device applications. Traditional personalization approaches often require labeled datasets and substantial computational resources. In contrast, ASLS leverages self-supervised learning for dynamic personalization, using a user profiling layer to gather interaction data and a neural adaptation layer for real-time model fine-tuning. This setup enables continuous learning from user feedback, aligning model outputs more closely with user-specific contexts while minimizing resource demands. Experimental results demonstrate ASLS’s effectiveness in enhancing user engagement and satisfaction, positioning it as a promising approach for creating responsive, context-aware on-device LLMs.

**Strengths:**

1. The paper focuses on a interesting and promising problem: On-Device LLM Personalization
2. The structure of the paper is reasonable

**Weaknesses:**

1. It's hard to understand the whole workflow because the paper lacks a workflow figure
2. Method is simple. Althought I approve simple but effective method, I can't get where is the "self-supervised learning" and can't understand why it is for "on-device LLM". This does not seem to be a method designed specifically for on-device LLM.

**Questions:**

See above

---

### Official Review · Reviewer_uQnh · 2024-11-04

**Soundness:** 1
**Presentation:** 1
**Contribution:** 1
**Rating:** 3
**Confidence:** 3

**Summary:**

This paper proposes an adaptive self-supervised learning strategy (ASLS), which consists of a user analysis layer for collecting interaction data and a neural adaptation layer for real-time model fine-tuning.

**Strengths:**

**S1.** The problem of "dynamically personalizing LLMs" is interesting and valuable.
**S2.** The authors claim that the method does not require extensive labeled data.

**Weaknesses:**

**W1.** Regarding the introduction section, it does not provide a clear motivation and the logic is somewhat unclear. For example, the fourth paragraph is very confusing, as it includes too many disparate elements—recommendation, multi-modal object recognition, and fairness.

**W2.** The connections between different parts of the method are also unclear, with issues of inconsistent notation. Additionally, there is no clear demonstration of how it better addresses the problem of dynamic updates; it still seems that updates are needed for each proposed module when new data appears.

**W3.** The experiments are also confusing. For example, in Table 1, why do different methods correspond to different datasets? How does this ensure fair comparison?

**W4.** It is not clear how the paper demonstrates self-supervised learning. According to equation (2), it still appears to be supervised learning.

**W5.** The selection of baselines may also need improvement. For instance, the chosen recommendation baseline, PALR, is not the current state-of-the-art LLM-based recommendation method.

**Questions:**

The writing in this paper needs improvement. Currently, I do not clearly understand how the method works, and the experimental setup is also confusing. See the details in the Weaknesses section.

---

### Official Review · Reviewer_vEb2 · 2024-11-04

**Soundness:** 2
**Presentation:** 1
**Contribution:** 1
**Rating:** 3
**Confidence:** 5

**Summary:**

The paper proposes a Self-Supervised Learning method to update LLMs by user interaction data for personalization on-device. The model contains two layers, a user layer to generate uer presentation by interaction data and a fine-tuning adaptation layer for dynamic modelling. The method is verified on various datesets.

**Strengths:**

1. The paper focus on decreasing computation as personalizing based on LLMs on-device.
 2. The paper conducts experiments on multiple datasets from various domains.

**Weaknesses:**

1. The definition of on-device personalization is not clear in the paper.
2. The metrics are not explained and defined, the reviewer cannot know what is the objective of this paper.
3. The comparison is not fair according to Table1 which compare various methods under different datasets, and no explain of such setting is provided.
4. The explain of layers of the method is vague, the review cannot follow due to lack of explanation and connection between section3.1-3.3

**Questions:**

In addition to the weakness:
1. what are the metrics of Eval Metric 1 Eval Metric 2 Eval Metric 3 Eval Metric 4 Eval Metric 5?
2. What's the definition of engagement score, satisfication rate and Importance Score in the evaluation?

---

### Official Review · Reviewer_PDiu · 2024-11-05

**Soundness:** 2
**Presentation:** 1
**Contribution:** 1
**Rating:** 3
**Confidence:** 4

**Summary:**

This paper introduces an Adaptive Self-Supervised Learning Strategy (ASLS) framework for dynamic, on-device personalization of large language models (LLMs). The framework includes two layers: a user profiling layer that gathers interaction data and a neural adaptation layer that fine-tunes the model in real-time based on this data, enabling responses tailored to specific user contexts.

**Strengths:**

1.	Studying on-device LLM personalization has significant practical application significance.
2.	The author claimed that the proposed method can achieve real-time model fine-tuning.

**Weaknesses:**

1. The quality of writing in the method section needs improvement. In particular, inconsistent use of symbols and lack of explanations make the methodology difficult to understand. It is recommended that the authors revise this section and include an illustrative diagram to improve readability.
2. Equation (2) indicates that the personalization approach in this paper still relies on label fitting, suggesting it has not eliminated the dependence on extensive labeled data as claimed in the introduction.
3. This paper's general approach looks like customizing personalized parameters for LLMs based on user interaction data, and this core idea seems quite similar to HYDRA [1]. What are the main innovations presented in this paper, and what are its advantages?
4. The readability of the experimental results and analysis is poor. The experimental tables are difficult to understand; for example, in Table 1, why do different methods correspond to different datasets? Regarding the analysis of experiments, the statement "Significant enhancements observed in user engagement metrics" does not provide a clear definition of what engagement metrics are.
5. There is a lack of comparison with some baselines in the experiments, such as [1].
[1] HYDRA: Model Factorization Framework for Black-Box LLM Personalization.

**Questions:**

See the Weaknesses section for the main questions. How does the proposed method ensure real-time model fine-tuning?

---

### Meta-Review · Area_Chair_bxCw · 2024-12-19

**Metareview:**

This paper proposes Adaptive Self-Supervised Learning Strategies (ASLS) for the dynamic, on-device personalization of large language models (LLMs). The authors claim that ASLS leverages self-supervised learning for real-time adaptation through a user profiling layer and a neural adaptation layer. The goal is to personalize LLMs for individual users while minimizing resource demands. The paper evaluates its method across multiple datasets, reporting improvements in user engagement and satisfaction.

The primary strength of this submission lies in addressing a timely and practically significant challenge—on-device LLM personalization. This problem aligns with emerging demands for privacy-preserving and resource-efficient AI. The authors claim to use innovative self-supervised techniques for real-time fine-tuning, and they attempt to validate their approach with experiments.

However, the weaknesses of the paper outweigh its strengths. First, there are critical issues in the clarity and rigor of the methodology. The method section is poorly written, with inconsistent notation, a lack of illustrative diagrams, and vague explanations of the proposed layers. Several reviewers noted confusion regarding how the self-supervised learning aspect was implemented and how it supports real-time on-device applications. Furthermore, the core contribution appears to be incremental and closely resembles prior work, such as the HYDRA framework, with insufficient clarity on the novel aspects of the method.

The experimental evaluation also raises significant concerns. The comparisons are not rigorous, with no clear rationale for using different datasets across baselines and no inclusion of state-of-the-art benchmarks. Additionally, the evaluation metrics, such as "engagement score" and "satisfaction rate," are not well-defined, leaving the results open to interpretation. Reviewers also criticized the experimental presentation, noting that tables were difficult to interpret and lacked explanation.

While the problem itself is promising, the paper falls short in presenting a robust and innovative solution. It lacks the theoretical clarity, methodological rigor, and experimental depth required for acceptance.

The primary reasons for recommending rejection are the lack of clarity and novelty in the proposed method, the insufficient and poorly explained experimental results, and the weak comparisons with relevant baselines. The authors need to address these issues comprehensively in a future submission to make their work competitive for publication.

**Additional Comments On Reviewer Discussion:**

During the rebuttal period, the authors attempted to address concerns raised by reviewers but provided limited clarification. Reviewer PDiu pointed out the similarity to prior work and asked for a clearer explanation of the method's novelty, which was not sufficiently addressed. Reviewer vEb2 highlighted the lack of metric definitions and experimental clarity. While the authors responded with additional explanations, the fundamental issues of unclear methodology and insufficient experimental justification remained unresolved. Reviewer uQnh's concerns about the experimental setup and baseline selection were also not adequately addressed.

The rebuttal did not provide compelling evidence to change the reviewers' initial assessments. The lack of clarity and rigor persisted in both the methodology and experiments, and the authors' responses failed to resolve the major concerns. As Area Chair, I weighed the reviewers’ assessments heavily, as they were consistent and well-supported.

Given these considerations, I concur with the reviewers' overall ratings and recommend rejecting this submission. The authors are encouraged to substantially revise their paper by addressing the methodological and experimental weaknesses and resubmitting to a future venue.

---

### Decision · Program_Chairs · 2025-01-22

Reject